# Unraveling the role of silicon in atmospheric aerosol secondary formation: A new conservative tracer for aerosol chemistry

Dawei Lu[1], Jihua Tan[2], Xuezhi Yang[1,2], Xu Sun[1], Qian Liu[1,2,3], Guibin Jiang[1,2]

[1]State Key Laboratory of Environmental Chemistry and Ecotoxicology, Research Center for Eco-Environmental Sciences, Chinese Academy of Sciences, Beijing 100085, China
[2]College of Resources and Environment, University of Chinese Academy of Sciences, Beijing 100049, China
[3]Institute of Environment and Health, Jianghan University, Wuhan 430056, China

*Correspondence to*: Qian Liu (qianliu@rcees.ac.cn)

**Abstract.** Aerosol particles are ubiquitous in the atmosphere and affect the quality of human life through their climatic and health effects. The formation and growth of aerosol particles involve extremely complex reactions and processes. Due to limited research tools, the sources and chemistry of aerosols are still not fully understood, and until now they are normally investigated by using chemical species of secondary aerosols (e.g., $NH_4^+$, $NO_3^-$, $SO_4^{2-}$, SOC) as tracers. Here we investigated the role of silicon (Si), a ubiquitous but relatively inert element, during the secondary aerosol formation process. We analyzed the correlation of Si in airborne fine particles ($PM_{2.5}$) collected in Beijing—a typical pollution region—with the secondary chemical species and secondary particle precursors (e.g., $SO_2$ and $NO_x$). The total mass of Si in $PM_{2.5}$ was found to be uncorrelated with the secondary aerosol formation process, which suggested that Si is a new conservative tracer for the amount of primary materials in $PM_{2.5}$, and can be used to estimate the relative amount of secondary and primary compounds in $PM_{2.5}$. This finding enables the accurate estimation of secondary aerosol contribution to $PM_{2.5}$ by using Si as a single tracer rather than normally used multiple chemical tracers. In addition, we show that the correlation analysis of secondary aerosols with the Si isotopic composition of $PM_{2.5}$ can further reveal the sources of the precursors of secondary aerosols. Therefore, Si may provide a new tool for aerosol chemistry studies.

## 1 Introduction

Atmospheric particulate pollution is a global environmental issue that seriously threatens human health and sustainable development. The fine particulate matter with size less than 2.5 μm ($PM_{2.5}$) is of greatest concern because it poses significant risks to human health and is a major cause of haze phenomenon (Pope et al., 2009; Pope et al., 2002). Understanding the sources and chemistry of aerosols is critical for air pollution control. The aerosol particles can be directly emitted from primary sources (primary particles) or secondarily formed from gaseous precursors (e.g., $SO_2$, $NO_x$, $NH_3$, and volatile organic compounds (VOCs)) through complex chemical reactions and processes in the atmosphere (secondary particles) (Zhang et al., 2012b; Zhang et al., 2015). Particularly, secondary aerosol (SA) is of great importance as it contributes to a large portion of particulate pollution in most pollution regions (e.g., China and India) (Huang et al., 2014). However, until now, the formation mechanism of secondary aerosols is not fully understood and it is difficult to accurately estimate secondary aerosols due to the extreme complexity of aerosol chemistry and limited research tools. The traditional method is based on the combined estimates of secondary inorganic aerosol (SIA) and secondary

organic aerosol (SOA) using multiple chemical components indicative of secondary chemistry, e.g. $NH_4^+$, $NO_3^-$, and $SO_4^{2-}$ for SIA (Bi et al., 2007), and organic carbon (OC) and elemental carbon (EC) for SOA (Docherty et al., 2008). However, the accuracy of the method is still controversial and measurements of these multiple tracers are laborious (Schmid et al., 2001; Watson et al., 2005).

As the second highest abundant element in the Earth's crust, silicon (Si) is ubiquitous in terrestrial systems including in atmospheric aerosols. The roles of Si in continental and marine environments have been well clarified in biogeochemical studies (Basile-Doelsch, 2006). However, as a mismatch of the ubiquitousness of Si, the research on atmospheric Si is rather limited (Bzdek et al., 2014; Lu et al., 2018). We note that Si is a special element compared with other high-abundance ones (e.g., O, C, N, and S). It is relatively inert that normally forms bonds only with oxygen (in $Si^{IV}$) and does not form volatile compounds in the natural environment (Savage et al., 2014). In fact, over 90% of Si in the Earth's crust is composed of nonvolatile silicate minerals that should not be able to serve as gaseous precursors for secondary aerosols (an exception is synthetic organosilicons that will be discussed later). Therefore, Si may provide a different route to aerosol chemistry from that using traditional tracers.

In this study, we aimed to clarify the role of Si during the secondary aerosol formation process in the atmosphere. Based on the nature of Si, we hypothesize that Si in $PM_{2.5}$ is unaffected by the secondary particle formation process and predominantly emitted from primary sources (as depicted in Fig. 1). Therefore, the particular properties of Si (i.e., high abundance and chemical inactiveness) may make it a new conservative tracer for aerosol chemistry studies. To specify this point, we herein show the estimation of secondary aerosols based on the dilution effect of Si during the secondary aerosol formation process by using Si as a single tracer and compare it with the traditional method that uses multiple chemical tracers.

## 2 Methodology

### 2.1 Chemicals and reagents

The standard reference material of urban atmospheric particulate matter (NIST-1648a) and the Si isotope standard (NIST SRM-8546) were purchased from the National Institute of Standards and Technology (Gaithersburg, MD). The Si isotope standard IRMM-017 was purchased from the Institute for Reference Materials and Measurements (GEEL, Belgium). The element calibration standard solution was from Agilent (Santa Clara, CA). Sodium hydroxide was from Beijing Ruikai Co. (Beijing, China). Nitric acid was from Merck (Darmstadt, Germany). Hydrochloric acid was from Beijing Chemicals Works (Beijing, China). Hydrogen peroxide was from Sinopharm Chemical Reagent Co. (Shanghai, China). Ultrapure water (18.3 MΩ·cm) obtained from a Milli-Q Gradient system (Millipore, Bedford) was used.

### 2.2 Sampling of $PM_{2.5}$ samples

The $PM_{2.5}$ and primary source samples were collected around the Beijing region, China, as described in a previous report (Lu et al., 2018). Briefly, the $PM_{2.5}$ samples were collected in an urban district of Beijing in 2013 by using a low-volume air sampler (Partisol 2025i, Thermo Fisher, USA) at a flow rate of 16.7 L/min. The sampling site was ca. 20 m height above the ground and surrounded by office and residential buildings

(116.4254 °E, 40.0481 °N). We selected the days with relatively high $PM_{2.5}$ concentrations in each week in 2013 ($n = 63$) to reflect the haze pollution condition of the whole year. Several haze events over consecutive days were also included for investigating the SA formation process. The sampling dates with the $PM_{2.5}$ concentrations and meteorological parameters are detailed in Table S1, Supplement. The $PM_{2.5}$ samples were collected onto Whatman 3-5 Teflon membrane filters (Ø = 47 mm, Maidstone, UK) or Munktell MK360 quartz filters (Ø = 90 mm, Maidstone, UK) and weighed by the giant gravimetric balance method (Yang et al., 2015). Different filters were chosen depending on target analytes, e.g., the analysis of Si and water soluble inorganic ions must be performed with Teflon filters, while quartz filters should be used for OC and EC analysis.

**2.3 Sample preparation procedures**

For the analysis of Si concentration and isotopic composition, the samples were digested and purified as reported previously (Georg et al., 2006; Zambardi and Poitrasson, 2011). Briefly, the sample was first dried in a silver crucible in a muffle furnace at 1000 K for 10 min. Then, high-purity solid NaOH was added to the sample at a ratio of 1:20 and the mixture was heated at 1000 K for another 10 min followed by cooling down to room temperature. The obtained fusion cake was dissolved with 2 mL of water followed by 24 h incubation, and then the solution was acidized by an HCl solution to pH ~2.

To eliminate the interference from matrix, the samples were purified by cation exchange column chromatography. The cation resin (Dowex 50WX8, 200-400 mesh) in $H^+$ form was packed to a 1.8 mL resin bed in a BioRad column and then rinsed with HCl, $HNO_3$ solution, and water until the eluate reached pH ~7. Afterwards, 1 mL of the sample solution was loaded to the resin and eluted with 2 mL of water. Because Si in the solution is present mostly in the form of non-ionic $Si(OH)_4$ and anionic $H_3SiO_4^-$ (Georg et al., 2006), cationic species could be removed from Si species by the cation exchange resin. The recovery during the column purification procedures with the IRMM-017 was > 95%. The recovery of Si during the whole sample preparation procedures with the NIST-1648 was 89.9-96.2%.

**2.4 Measurement of Si concentration**

The concentration of Si was measured on an Agilent 8800 inductively coupled plasma mass spectrometer (Santa Clara, CA, USA). The clean blank membrane filters have also been analyzed using the same procedures to subtract the background signals from the samples.

**2.5 Estimation of secondary aerosols in $PM_{2.5}$ by using the Si-dilution method**

Providing that the total Si in $PM_{2.5}$ ($Si_{PM2.5}$) keeps unchanged during the secondary growth of $PM_{2.5}$, we can obtain

$$m_{PM2.5} = m_{pri} + m_{sec} \tag{1}$$

$$Si_{PM2.5} = m_{PM2.5} \times C_{PM2.5} = m_{pri} \times C_{pri} \tag{2}$$

where $m_{PM2.5}$, $m_{pri}$, and $m_{sec}$ represent the mass of $PM_{2.5}$, primary particles, and secondary particles, respectively; $Si_{PM2.5}$ represent the total Si mass in $PM_{2.5}$; $C_{PM2.5}$ and $C_{pri}$ represent Si abundance in $PM_{2.5}$ and primary particles, respectively. Thus, the contribution of secondary aerosols to $PM_{2.5}$ ($f_{sec}$) can be estimated by the Eq. (3):

$$f_{\text{sec}} = \frac{m_{\text{sec}}}{m_{\text{PM2.5}}} = 1 - \frac{C_{\text{PM2.5}}}{C_{\text{pri}}} \qquad\qquad (3)$$

The $C_{\text{PM2.5}}$ can be obtained by direct measurement of the collected PM$_{2.5}$ samples, and the $C_{\text{pri}}$ can be estimated from the mixed Si abundances of primary sources:

$$C_{\text{pri}} = \sum(C_i \times f_i) \qquad\qquad (4)$$

where $C_i$ and $f_i$ represent the Si abundance and mass fraction of different primary sources ($f_i = m_i / \sum m_i$). In Eq. (4), the $C_i$ can be obtained by measuring the collected primary source samples. In order to obtain $f_i$, we need to know the emission amount ($m_i$) of each source. The $m_i$ of anthropogenic sources in the Beijing region was adopted from the Multi-resolution Emission Inventory for China (MEIC) database (MEIC; Li et al., 2017). For the $m_i$ of natural sources (i.e., dusts), it can be calculated based on the isotopic mass balance of Si in PM$_{2.5}$:

$$\delta^{30}\text{Si}_{\text{PM2.5}} = \delta^{30}\text{Si}_{\text{pri}} = \sum(\delta^{30}\text{Si}_i \times C_i \times f_i)/C_{\text{pri}} \qquad\qquad (5)$$

where $\delta^{30}\text{Si}_{\text{PM2.5}}$ represents Si isotopic composition of PM$_{2.5}$ and $\delta^{30}\text{Si}_i$ represents Si isotopic signatures of different primary sources. The $m_i$ values for different primary sources used in the calculation are given in Table 1.

**2.6 Uncertainty analysis in the estimation of secondary aerosols**

The uncertainty of secondary aerosol contribution estimated by the Si-dilution method was obtained by the error propagation calculation. The general formula for the error propagation calculation is as follows (Ku, 1966; Larsen et al., 2012):

$$S_f = \sqrt{\sum\left(\frac{\partial f}{\partial x_i} \times S_i\right)^2} \qquad\qquad (6)$$

where $S_f$ represents the standard deviation of the function $f$, $x_i$ represents the variables in the function $f$, $S_i$ represents the standard deviation of $x_i$. The Eq. (6) was applied to each step of the calculation of secondary aerosol contribution ($f_{\text{sec}}$) to obtain the uncertainty of the final result. Briefly, the uncertainty of $f_{\text{sec}}$ is dependent on several variables in the calculation, including the measured Si abundance in PM$_{2.5}$ ($C_{\text{PM2.5}}$), Si abundance ($C_i$) and mass fraction of primary sources ($f_i$), and Si isotopic composition of PM$_{2.5}$ ($\delta^{30}\text{Si}_{\text{PM2.5}}$) and primary sources ($\delta^{30}\text{Si}_i$). The errors of $C_{\text{PM2.5}}$, $C_i$, $\delta^{30}\text{Si}_{\text{PM2.5}}$, and $\delta^{30}\text{Si}_i$ can be directly obtained in the experimental measurements. The $f_i$ is given by the MEIC database (MEIC; Li et al., 2017). Note that the MEIC is a public emission inventory database that has been widely applied and well validated in air pollution research in China (Jiang et al., 2015; Zheng et al., 2015; Guan et al., 2014; Lin et al., 2014; Zhang et al., 2012a), but it does not include error information for the data. So, the error of $f_i$ was not included in the calculation. In this way, the uncertainty of the annual mean $f_{\text{sec}}$ in 2013 was calculated to be 26.1%. This value included both the method uncertainty and the variations on different dates. Nevertheless, it should be noted that emission inventory actually affected the uncertainty of the estimate result. For example, if the uncertainty of the emission inventory was assumed to be 5%, the uncertainty of the annual mean $f_{\text{sec}}$ in 2013 would be increased to 29.3%. On the other hand, the MEIC database used here can only provide yearly emission mass data, so using a high temporally resolved emission inventory may further increase the accuracy of the result.

**3 Results and discussion**

**3.1 Correlation of Si with the secondary species in PM$_{2.5}$**

To verify the role of Si as depicted in Fig. 1, we analyzed Si abundance and secondary aerosol tracers (e.g., NH$_4^+$, NO$_3^-$, SO$_4^{2-}$, OC, and EC) in PM$_{2.5}$ samples collected around Beijing, China, a typical particulate pollution region, on haze days ($n = 63$) in 2013, in which year the particulate pollution in this region reached an unprecedentedly high level (Huang et al., 2014). We found that the PM$_{2.5}$ concentration showed a clear seasonal trend (higher in spring/winter than in summer/autumn) within the range of 21.7-337.1 μg/m$^3$ (Fig. S1 and Table S1, Supplement). The PM$_{2.5}$ profile of the year 2013 was consistent with that reported previously (Lu et al., 2018), confirming the representative of the data. All secondary species in PM$_{2.5}$ showed a similar seasonal trend with the PM$_{2.5}$ concentration (Fig. S1, Supplement). However, the total Si in PM$_{2.5}$ (expressed as Si$_{PM2.5}$ relative to air volume in μg/m$^3$) did not show any seasonal trends (Fig. S2, Supplement).

Furthermore, Fig. 2 shows the correlation of PM$_{2.5}$ and its Si content with the secondary species in PM$_{2.5}$. It can be seen that the PM$_{2.5}$ concentration was highly linearly correlated with the secondary species ($P < 0.01$). These chemical species are directly indicative of secondary chemistry: sulfate is mainly converted from atmospheric SO$_2$ primarily emitted from coal combustion (Seinfeld and Pandis, 2006), nitrate originates from NO$_x$ emitted mainly from vehicle exhaust and power plants (Seinfeld and Pandis, 2006), and secondary organic carbon (SOC), as an indicator of SOA, derives from complex gaseous precursors (Hallquist et al., 2009). Specifically, PM$_{2.5}$ showed a higher correlation with SIA ($P < 0.001$) than with SOA ($P < 0.01$) probably due to the higher contribution of SIA to SA than SOA (Fig. S3, Supplement). However, as a stark contrast to the significant correlation in Fig. 2a-d, the Si$_{PM2.5}$ showed no significant correlation with any of the secondary species ($P > 0.14$); either, no obvious trend in Si$_{PM2.5}$ was observed with the increase of secondary species. These results strongly evidenced that the Si was inactive during the secondary chemical process of aerosols.

**3.2 Correlation of Si with the secondary precursors and relative humidity**

To further demonstrate the role of Si in secondary aerosols, we analyzed the correlation of Si$_{PM2.5}$ with the secondary precursors (SO$_2$ and NO$_x$) and relative humidity (RH). RH is an important meteorological parameter that affects the secondary aerosol formation (Tang et al., 2016). As shown in Fig. 3a-c, PM$_{2.5}$ was positively correlated with the atmospheric concentration of SO$_2$ and NO$_x$ ($P < 0.001$) and RH ($P < 0.05$). Noteworthily, all PM$_{2.5}$ samples yielded an OC/EC ratio $> 2$ (Fig. 3d), which indicated the generation of secondary aerosols (Chow et al., 1994). While for Si$_{PM2.5}$, it was not significantly correlated with any of the secondary precursors or RH ($P > 0.3$; Fig. 3e-g), again demonstrating that the Si$_{PM2.5}$ was not affected by the secondary chemistry of aerosols. Furthermore, we also found that the Si$_{PM2.5}$ was not correlated with the PM$_{2.5}$ concentration (Fig. S4, Supplement).

**3.3 Effects of synthetic organosilicons and dry deposition**

Based on the aforementioned results, we infer that Si should mainly be present in primary particles or act as nuclei during the formation of secondary particles. This was consistent with the non-volatile nature of most Si-containing compounds in the nature environment. An exception is synthetic low-mass organic Si compounds

(e.g., methylsiloxanes (Schweigkofler and Niessner, 1999; Ahrens et al., 2014; Xu et al., 2012)) that may be able to transform into aerosol nanoparticles via hydroxyl radicals-mediated oxidative cleavage of Si-C bonds in the upper atmosphere (Atkinson, 1991; Sommerlade et al., 1993; Atkinson et al., 1995; Wu and Johnston, 2017). However, previous studies on the impact of low-mass organosilicons on the air quality have demonstrated that organosilicons did not contribute to the lower atmosphere aerosol formation (Graiver et al., 2003; Sommerlade et al., 1993). As a result, the U.S. Environmental Protection Agency (EPA) has excluded organosilicons from the regulation concerning restriction of VOCs in the atmosphere. Recently, Janechek et al. found that the oxidative product concentration of cyclic volatile methylsiloxanes was ~10-40 times lower than their parent compounds in the U.S. (Janechek et al., 2017). While, another study by Wu and Johnston reported that secondary aerosol yields of siloxane oxidation might reach 15% (Wu and Johnston, 2017), suggesting that the contribution of organosilicons to secondary aerosols might vary in different regions. Thus, it may be necessary to consider the secondary Si production to determine whether it should be included in the secondary aerosol calculation. Note that the secondary Si production depends on the atmospheric concentration of both organosilicons and hydroxyl radicals. Up to now, no data about atmospheric organosilicons in the studied region was reported. Despite that, some recent studies reported the measurement and modeling results of atmospheric hydroxyl radicals in the North China Plain (Tan et al., 2017; Tham et al., 2016). The daily maximum concentration of atmospheric hydroxyl radicals was in the range of $(5 - 15) \times 10^6$ cm$^{-3}$ (~$10^{-10}$ mol m$^{-3}$). Since hydroxyl radical is a crucial condition for the siloxane oxidation, the extremely low concentration of hydroxyl radicals in the atmosphere could greatly limit the secondary Si production from siloxanes in the studied region, suggesting that the secondary Si production could be negligible compared with the high-abundance Si in primary particles. This deduction has also been experimentally verified by the irrelevance of Si$_{PM2.5}$ with the secondary chemistry of aerosols in Figs. 2 and 3.

We have also considered the influence of dry deposition on the Si abundance in PM$_{2.5}$. The dry deposition is a size-dependent process that can cause coarse particles to fall more quickly than fine particles. However, it was found that the average Si abundance did not change significantly when the particle size was smaller than 2.5 μm (Tan et al., 2016). That is to say, for fine particles (< 2.5 μm), the dry deposition rate would not significantly affect the Si abundance. Thus, the Si-dilution effect with PM$_{2.5}$ should be predominantly controlled by the secondary aerosol formation.

**3.4 Secondary aerosol estimation using the Si-dilution method**

The special role of Si in PM$_{2.5}$ enables us to gain insights into the secondary particles in PM$_{2.5}$ by using Si as a new conservative tracer. Here, as an example, we show the estimation of secondary aerosol contribution to PM$_{2.5}$, which was thought as a complicated and difficult task. Providing that the Si$_{PM2.5}$ keeps unchanged during the secondary growth of PM$_{2.5}$, the secondary formation can cause a dilution of the Si abundance in PM$_{2.5}$. Thus, it is possible to estimate the secondary aerosol contribution to PM$_{2.5}$ ($f_{sec}$) simply by the dilution factor (see Eq. (3) in the Sect. 2.5). To this end, the theoretical Si abundance in primary particles ($C_{pri}$) and the final Si abundance in PM$_{2.5}$ ($C_{PM2.5}$) need to be known. The $C_{PM2.5}$ can be directly measured with the collected PM$_{2.5}$ samples. In this study, an annual mean value of 1.56% ($n = 63$) was obtained. For the $C_{pri}$, it can derive from the mixed Si abundance of primary sources based on individual Si abundance and emission amounts of primary sources: the

former can be measured with the collected primary source samples, and the latter can be given by a public emission inventory (e.g., MEIC database). It should be noted that the emission inventory does not include the emission amount of natural primary sources (i.e., dusts) (MEIC), which can be estimated by the isotopic mass balance of Si in $PM_{2.5}$ based on the emission data of other sources and Si isotopic signatures of primary sources (see Table 1) (Lu et al., 2018). The whole estimation procedures are shown in Fig. 4a. In this way, the theoretical mean Si abundance in primary particles was calculated to be 7.51%. That is, the theoretical Si abundance in primary particles (7.51%) was diluted to be 1.56% by the secondary aerosol formation. Based on these values, the mean contribution of secondary particles to $PM_{2.5}$ on haze days in 2013 in the studied region was easily calculated to be 79.2% (Fig. 4b). In the same way, we have also estimated the secondary aerosol contribution to $PM_{2.5}$ in different seasons (Fig. 4c).

Note that the MEIC database only provides yearly emission data of primary sources. Thus, using a higher temporally resolved emission inventory in future studies may further increase the accuracy of the result. In addition, the inter-regional atmospheric transport of aerosols may also affect the estimate result. Here the $PM_{2.5}$ was assumed to mainly derive from local emission of primary particles and SA precursors. In fact, a portion of it might also come from adjacent cities/regions via atmospheric transport, which could cause uncertainties to the result. Therefore, using primary source information and emission inventory covering adjacent cities/regions in the calculation may also improve the accuracy of the result.

On the other hand, VOCs including siloxanes are likely to become increasingly significant in the SA formation as aerosols originating from fossil fuels may become less important over time (McDonald et al., 2018). The method present here is based on the dilution effect of Si in primary particles during the SA formation, and thereby it should be suitable for the environment where secondary Si production is minimal (e.g., in heavily polluted urban regions with a large fossil fuel contribution like Beijing). Nevertheless, the method also keeps the flexibility of being modified to be applicable for the cases when the secondary Si production is not negligible. In that case, the secondary Si production mass needs to be estimated and included in the mass balance calculations (Sect. 2.5).

### 3.5 Uncertainty analysis and comparison with the traditional method

The uncertainty of the estimate result could be obtained by the error propagation calculation (Ku, 1966; Larsen et al., 2012) from the errors of variables used in the calculation (see the Sect. 2.6 for details). In this way, the uncertainty of the annual mean SA contribution in 2013 was calculated to be 26.1%. This value included the method uncertainty and the variations on different dates but did not include the effect of emission inventory because the MEIC database does not include error information for the data (see the Sect. 2.6). Thus, the effect of the uncertainty of emission inventory should be borne in mind when understanding the uncertainty of the result. As a comparison, the traditional method uses more variables (e.g., $NH_4^+$, $NO_3^-$, $SO_4^{2-}$, EC, and OC) in the calculation, which may bring in large uncertainties to the result. For example, Schmid et al. reported that the RSD in the EC measurement could reach 36.6-45.5% (Schmid et al., 2001), which would cause large uncertainties in calculating the ratio of OC/EC and consequently cause the SOC to be biased by up to 64% (Guo et al., 2014). Meanwhile, the approach to estimate SOA by multiplying the SOC by an empirical coefficient is still controversial (Docherty et al., 2008). However, it is difficult to make a direct comparison on uncertainties

between the Si-dilution method and traditional multi-tracer method due to the absence of error information of emission inventory. Despite that, considering that the present method uses only a single tracer, it seems easier to control the uncertainties with the present method than with the traditional multi-tracer method.

We also compared the estimate results of the Si-dilution method with the traditional multi-tracer method (Table S2, Supplement). As shown in Fig. 4b-c, generally, the SA contribution estimated by the Si-dilution method was slightly higher than that obtained by the traditional method. Such a difference can be explained by the different strategies of the methods: the traditional method is based on limited rather than all active secondary species in $PM_{2.5}$, and thus the SA may be prone to be underestimated; while, the Si-dilution method only uses an inactive species (i.e., Si), and possible loss of Si-containing components during the sample pretreatment procedures may cause the result to be overestimated. Despite that, the results between the two methods were close. These two methods were further compared with a special haze episode during Oct 16 to Oct 17 in 2013 in Beijing. During this episode, the $PM_{2.5}$ concentration burst from 26.1 to 197.8 μg/m$^3$ (Fig. 4d). The Si-dilution method showed that the SA contribution increased remarkably from 35.6% for Oct 16 to 75.0% for Oct 17, which was highly consistent with that obtained with the traditional method (Fig. 4e). This could also verify the accuracy of the Si-dilution method.

**3.6 Sources of secondary particle precursors revealed by Si isotopic signatures**

In addition to the SA formation estimation, the correlation analysis of Si isotopic composition of $PM_{2.5}$ with the SA can further reveal the sources of the precursors of SA. Note that the Si isotopic composition is independent of Si abundance of $PM_{2.5}$, and it can actually reflect the primary sources of $PM_{2.5}$, because different primary sources have different Si isotopic signatures (e.g., coal burning and industrial emission are $^{30}$Si-depleted sources, while vehicle emission and dusts are $^{30}$Si-enriched sources) (Lu et al., 2018). From Fig. S5, Supplement, the secondary aerosols were negatively correlated with the $\delta^{30}$Si of $PM_{2.5}$, suggesting that $^{30}$Si-depleted primary sources (e.g., coal burning and industrial emission) contributed more precursors to secondary aerosols than $^{30}$Si-enriched sources. Specifically, from Fig. 5a, the $\delta^{30}$Si showed a decline trend with the increase in $SO_4^{2-}$ concentration ($P = 0.006$), suggesting that the $^{30}$Si-depleted primary sources contributed more precursor (i.e., $SO_2$) to the $SO_4^{2-}$ species than $^{30}$Si-enriched sources. This was consistent with the fact that $SO_2$ mainly derives from coal burning that is a $^{30}$Si-depleted source (Lu et al., 2018). The $NH_4^+$ and SOC concentration was also negatively correlated with the $\delta^{30}$Si (Fig. 5c and 5d), suggesting that the precursors of these secondary species were also largely contributed by $^{30}$Si-depleted primary sources. However, the $NO_3^-$ concentration did not show any correlation with the $\delta^{30}$Si ($P = 0.63$; Fig. 5b). This could be explained by the fact that the sources of $NO_3^-$ were more uncertain and involved both $^{30}$Si-enriched (e.g., vehicle exhaust) and $^{30}$Si-depleted primary sources (e.g., power plants) (Seinfeld and Pandis, 2006).

To further show the correlation of Si isotopic signatures with the SA precursors, we also analyzed two typical haze episodes in 2013 in Beijing (Fig. S6, Supplement). In the episode during Nov 30 to Dec 1, 2013 (Fig. S6a-j, Supplement), the $PM_{2.5}$ rapidly increased from 26.1 to 197.8 μg/m$^3$, during which the SIA, SOA, and their major constituents (i.e., $NH_4^+$, $NO_3^-$, $SO_4^{2-}$, EC, and SOC) synchronously increased and the SA precursors ($NO_x$ and $SO_2$) also showed a rapid rise. Meanwhile, the $\delta^{30}$Si of $PM_{2.5}$ shifted negatively with the SA formation, suggesting that the SA precursors were largely contributed by $^{30}$Si-depleted sources. In another haze episode

during Dec 6 to Dec 7 in 2013 (Fig. S6k-t, Supplement), similar phenomena were observed that the $\delta^{30}$Si of PM$_{2.5}$ shifted negatively with the SA formation. Thus, all results mentioned above suggested that $^{30}$Si-depleted primary sources (e.g., coal burning and industrial emission) contributed more SA precursors than $^{30}$Si-enriched sources. This deduction was consistent with previous knowledge that coal burning and industrial emission contributed more precursors to SA (e.g., SO$_2$, NO$_x$, and VOCs) (Sun et al., 2016). Accordingly, we show that the Si isotopic composition not only indicates the primary sources, but also can reveal the sources of precursors of secondary species of PM$_{2.5}$.

### 3.7 Implications for air pollution control policies

The SA contribution obtained here (79.2 ± 26.1% for the entire 2013 and 88.7 ± 8.9% for Jan 2013; Fig. 4) was overall higher than that in a previous report (30-77% for four cities in China in Jan 2013) due probably to the different sampling dates (Huang et al., 2014). Noteworthily, these results have been verified by two independent methods (Fig. 4). Thus, more strict policies should be enforced to reduce the emission of secondary particle precursors including SO$_2$, NO$_x$, and VOCs for haze pollution control, especially for NO$_x$ and VOCs which have not aroused full attention in current pollution control strategies (Wang et al., 2013). Furthermore, the $^{30}$Si-depleted primary sources (e.g., coal burning and industrial emission) should be priorly regulated because they make a great contribution to both primary particles and secondary particle precursors.

### 4 Conclusions

In summary, we have investigated the role of Si during aerosol secondary formation and show that the high abundance and chemical inactiveness of Si make it a new conservative tracer in investigating the aerosol formation process. Based on the Si-dilution effect, we have proposed a new method to estimate the secondary aerosol contribution to PM$_{2.5}$ by using Si as a single tracer. In addition, the sources of the precursors of secondary aerosols can be revealed by the correlation analysis of secondary aerosols with the Si isotopic composition of PM$_{2.5}$. Overall, this study not only enriches our understanding on the role of Si during the aerosol formation process, but also adds a new alternative tool into the toolbox of aerosol chemistry research. However, it should be noted that this method is expected to be useful only in regions where crustal particles represent a measurable amount of PM$_{2.5}$, and in areas dominated by biogenic aerosols (e.g., forests) its usefulness is still questionable. Meanwhile, some measures may further improve the accuracy of the method, such as increasing the size and representativeness of primary source sample sets. Considering the inter-regional transport of PM$_{2.5}$, future efforts will also be made to include primary sources and emission inventory of adjacent cities/regions in the calculation.

### Supplement

The Supplement related to this article is available online at https://doi.org/XXXXXX.

**Author contributions**

QL designed the research; GJ supervised the project; DL performed most of experiments; JT measured the concentration of secondary species and calculated the secondary aerosol contribution using the traditional method; XY helped with the measurements; XS measured the atmospheric concentration of $SO_2$ and $NO_x$ and RH; QL and DL analyzed the data and wrote the paper.

**Competing interests**

The authors declare that they have no conflict of interest.

**Acknowledgements**

This work was financially supported by the National Natural Science Foundation of China (No. 21825403, 91843301, 91543104), the Chinese Academy of Sciences (XDB14010400, QYZDB-SSW-DQC018), and the National Basic Research Program of China (2015CB931903, 2015CB932003).

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

**Tables**

**Table 1. Parameters for primary sources used in the estimation of secondary aerosol contribution to PM$_{2.5}$.**

| | Coal combustion | Industrial emission | Biomass burning | Vehicle exhaust | Dusts |
|---|---|---|---|---|---|
| Emission mass ($m_i$; Mg/year) | 43813[a] | 34111[a] | 27476[b] | 5379[a] | 65199[c] |
| Si abundance ($C_i$)[d] | 8.06% | 0.79% | 1.01% | 0.2% | 14% |
| Si isotopic signature ($\delta^{30}Si_i$)[d] | -2.4‰ | -1.4‰ | -0.50‰ | 1.0‰ | -0.28‰ |

[a] Data from the MEIC database (MEIC).

[b] Since the MEIC database does not include data for biomass burning, the $m_i$ of biomass burning was estimated from the data of other sources according to source apportionment results of 2013 reported previously (Tian et al., 2016).

[c] In the calculation, soil, construction, and urban fugitive dusts are treated as a single source considering their similarity in Si abundance and Si isotopic signatures (Lu et al., 2018). The $m_i$ of dusts was calculated by using isotopic mass balance of Si in PM$_{2.5}$ (see Methodology).

[d] The Si abundance and natural Si isotopic signatures of primary sources around Beijing are adopted from a previous study (Lu et al., 2018).

**Figure captions**

**Figure 1.** Scheme showing the role of Si during the secondary formation process of aerosol particles. *Note:* this scheme only reflects the role of Si but does not show all reactions involved in the secondary aerosol formation.

**Figure 2.** Correlation analysis of $PM_{2.5}$ concentration and its Si content ($Si_{PM2.5}$) with the secondary species in $PM_{2.5}$. (**a-d**), Correlation of $PM_{2.5}$ concentration with $SO_4^{2-}$ (**a**), $NO_3^-$ (**b**), $NH_4^+$ (**c**), and SOC (**d**). (**e-h**), Correlation of $Si_{PM2.5}$ with $SO_4^{2-}$ (**e**), $NO_3^-$ (**f**), $NH_4^+$ (**g**), and SOC (**h**).

**Figure 3.** Correlation analysis of $PM_{2.5}$ concentration and $Si_{PM2.5}$ with typical secondary aerosol precursors and relative humidity (RH). (**a-d**), Correlation of $PM_{2.5}$ concentration with $SO_2$ (**a**), $NO_x$ (**b**), RH (**c**) and OC/EC ratio (**d**). (**d-f**), Correlation of $Si_{PM2.5}$ with $SO_2$ (**e**), $NO_x$ (**f**), and RH (**g**) and OC/EC ratio (**h**). The red dotted lines in (**d**) and (**h**) represent OC/EC = 2, which can be used to indicate the formation of secondary aerosols at OC/EC > 2 (Chow et al., 1994).

**Figure 4.** Comparison of the SA estimate results between the Si-dilution method and the traditional method. (**a**), A flow chart showing how the $f_{sec}$ was obtained by the Si-dilution method. The boxes come from estimations from emission inventory are labeled with asterisk. (**b**), Annual mean SA contribution to $PM_{2.5}$ in 2013 in Beijing. (**c**), Seasonal mean SA contribution in 2013 in Beijing. The error bars for the Si-dilution method include both method uncertainty and the variations of samples on different dates (see Sect. 2.6); while for the traditional method the error bars only represent the variations of samples on different dates (without method uncertainty). Please see text for a more detailed discussion on the uncertainties. (**d-e**) Analysis of SA contribution during a special haze episode. (**d**), Monitoring of $PM_{2.5}$ concentration variations during Oct 16, 2013 to Oct 17, 2013. (**e**), Daily SA contributions during the episode estimated by using the two different methods.

**Figure 5.** Correlation analysis of Si isotopic composition ($\delta^{30}Si$) of $PM_{2.5}$ with the secondary species ($SO_4^{2-}$ (**a**), $NO_3^-$ (**b**), $NH_4^+$ (**c**), and SOC (**d**)).

**Figures**

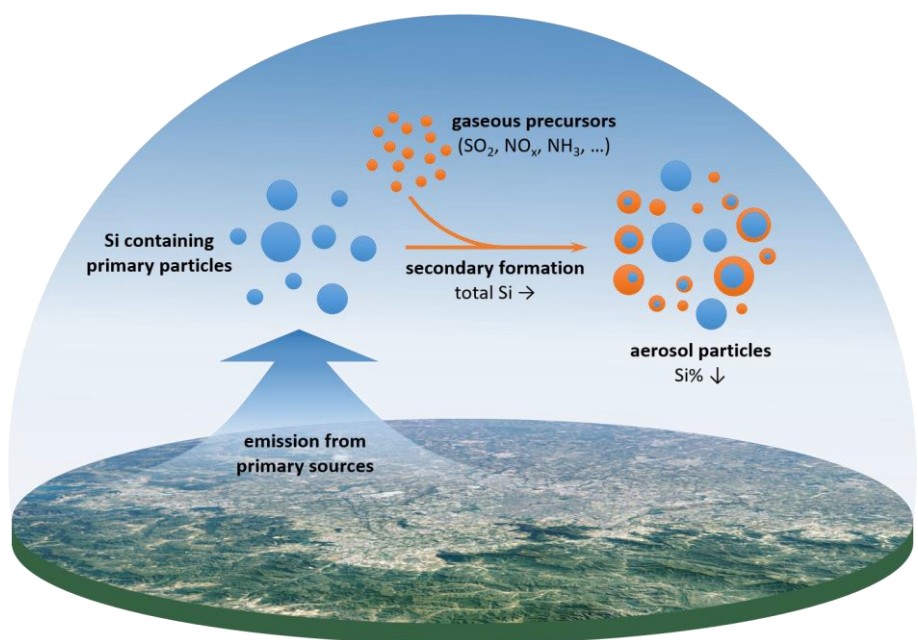

Figure 1

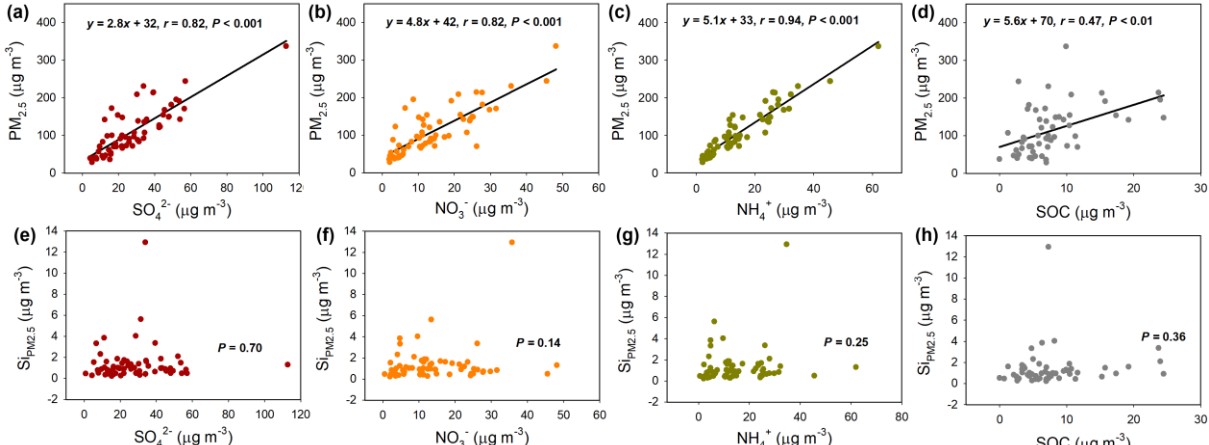

Figure 2

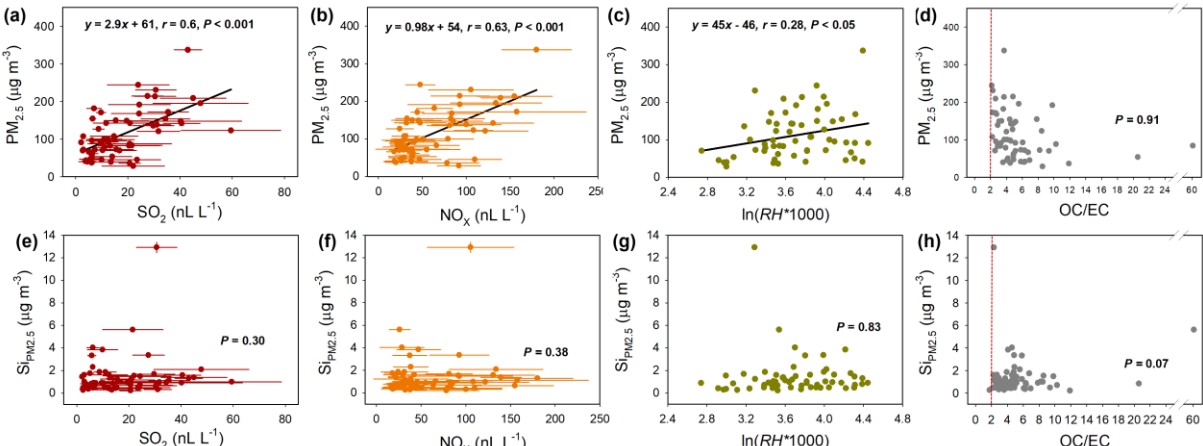

Figure 3

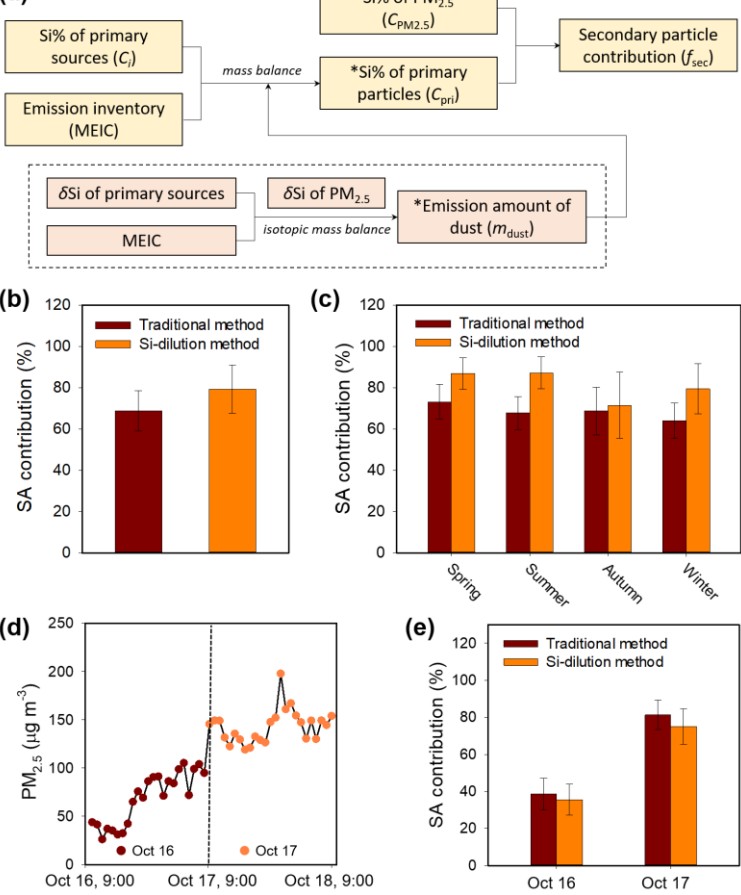

Figure 4

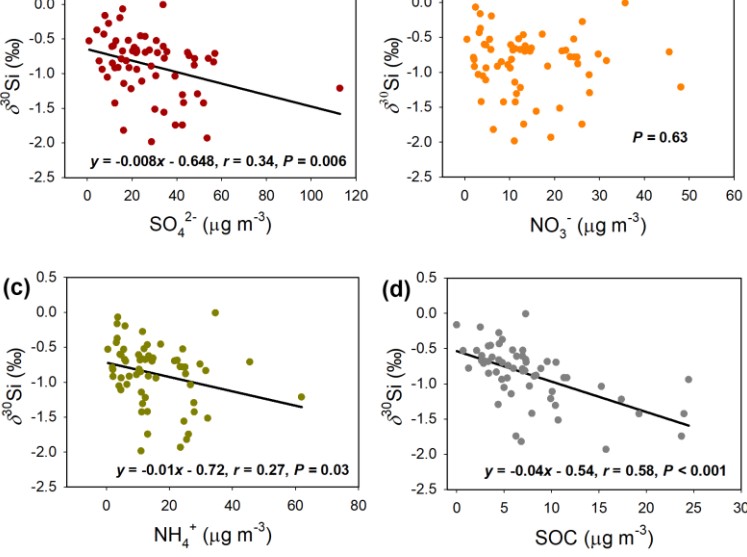

Figure 5