# Peer review of "Unraveling the role of silicon in atmospheric aerosol secondary formation: A new conservative tracer for aerosol chemistry"

_Atmospheric Chemistry and Physics, 2018_

## Referee Comment (RC1) · Anonymous Referee #1 · 5 Dec 2018

**Comments on acp-2018-914 "Unraveling the role of silicon in atmospheric aerosol secondary formation: A new conservative tracer for aerosol chemistry"**

**General comments:** In the Earth's crust, silicon (Si) is ubiquitous and relatively inert element. The Si also existed in atmospheric aerosols. In this article, the author reports the role of silicon (Si) in the secondary aerosol formation. They found uncorrelated of the total mass of Si in PM2.5 with the SA formation. Therefore, they suggested Si as a new conservative tracer for estimating the SA contribution to PM2.5. Furthermore, the author also correlated the Si isotopic signatures with $SO_4^{2-}$, $NO^{3-}$, $NH_4^+$, and SOC, then deduced the sources of these secondary components. This is a new method in estimating the secondary aerosol contribution to PM2.5 by using the Si as a tracer. It's meaningful for the atmospheric chemistry. However, there still some comments need to be solved before the manuscript can be published. The author should consider that chemical species of secondary aerosols (e.g., $NH_4^+$, $NO^{3-}$, $SO_4^{2-}$, SOC) as tracers are able to get the detail information of SA compare to use Si as a single tracer, what is the superiority of this method?

**Detail comments:**

**2.2 Sampling of PM2.5 samples**

(1) "the PM2.5 samples were collected around Beijing on random haze days (n = 100) in 2013". The author should give more detail information about the samples, such as the concentrations of PM2.5, meteorological parameters, etc. These environmental conditions can help the reader to understand the latter results and discussion more clearly. These are also very important for the conclusion especially the contribution of SA in PM2.5 during this article.

(2) What is the meaning of "random haze days (n=100)"? Can these samples represent the secondary formation of particles?

(3) The sampling sites should be given a brief introduction. For example, what the feature around the sampling sites? What the location of the sites?

**2.6 Uncertainty analysis in the estimation of secondary aerosols**

(1) Emission Inventory may introduce significant uncertainties because of uncertainties in activity-related data and emission factors. At least, the authors should state clearly the effect of emission inventory so that the readers can judge by themselves. Furthermore, "assuming that the relative emission ratios of primary sources are steady" what's the meaning? The high temporally resolved emission inventory could be used in Table 1, other than the total emissions mass. For example, the monthly emission mass from emission inventory could be used other than yearly emission mass.

**From 3.1 to 3.2, the author correlated the Si with the secondary species in PM2.5, the secondary precursors, and relative humidity. There are some comments about this part:**

(1) In the part of Methodology, the author stated that random haze days n=100, in this part, n=63. What the meaning of "random haze days"? why these days were used to do the analysis? The author found the PM2.5 concentration showed a clear seasonal trend. How was the representative of the data in the article? Have the author compared their results with references?

(2) All the samples were collected during haze days, why they don't collect some samples during clean days before these haze days? The comparison from clean days to haze days can help to give more information during the formation of SA.

**3.4 Secondary aerosol estimate using the Si-dilution method:**

(1) The uncertain caused by atmospheric transport should be considered in this study.

(2) Some special episodes should be analysis to support the views and better understand for the readers in this work.

**3.5 Comparison with the traditional method**

The author stated that the "traditional method bring in huge uncertainties to the result". Then the author compared the Si-dilution method with the traditional method, "the results between the two methods were actually very close". Why?

**3.6 Sources of secondary particle precursors revealed by Si isotopic signatures**

The discussion in section 3.6 is too simple and less persuasive. The author should give more detail analysis of even one haze case process. Furthermore, there are many works

have been done about the pollution case in the year 2013 in Beijing. The author should add more comparison between their results with the references.

**3.7 Implications for air pollution control policies**

Huang's results were based on the observation during the high pollution events of 5–25 January 2013 at the urban sites of Beijing. The SA contribution obtained in this article was 88.7% for Jan 2013. This result deduced from just about five days daily data (6/1/2013, 11/1/2013, 18/1/2013, 25/1/2013, 29/1/2013), can these data have the representative for 20 days from 5-25 Jan 2013? The SA contribution obtained was 79.2%, what's the uncertainty of these results?

Besides, there are some grammar and spelling mistakes. The format of some reference should be checked, such as line10 in section 3.3, Line12-13 in section 3.3.

---

## Referee Comment (RC2) · Anonymous Referee #2 · 7 Dec 2018

This manuscript (acp-2018-914) describes the use of Si to estimate the contribution of secondary aerosol to PM2.5 in the urban environment in Beijing, China. The approach relies on Si content from PM2.5 arising from primary sources only. Secondary aerosol concentrations are estimated by the inferred dilution of expected Si mass. This dilution is calculated by comparing measured Si concentrations in PM2.5 to estimated values based on sources of primary emission. This approach is compared to another approach where concentrations of several secondary species (NH4+, NO3-, SO2-2, secondary organic carbon) are used as tracers for secondary aerosol production. The Si dilution approach gives a value in line but slightly higher than those given by the approach using chemical components associated with secondary aerosols. The authors
argue the Si approach is more direct and has a lower uncertainty associated with it compared to the other approach. Moreover, the authors correlate secondary aerosol components in PM2.5 ($SO_4^{-2}$, $NO_3^-$, $NH_4^+$, and secondary organic carbon) with Si isotopic composition and observe some correlations that suggest some sources for secondary aerosol.

This manuscript represents a new approach to estimate secondary aerosol formation and is within the scope of Atmospheric Chemistry and Physics. However, several important issues, detailed below, must be carefully addressed by the authors before this manuscript is suitable for publication.

Comments:

1. The authors examined PM2.5 collected on "random haze days" (see page 3, line 3; page 5, line 4). It is unclear what "random haze day" means. Were the analysis days randomly selected? Were the selected days the ones with the highest PM2.5 concentrations, or days where PM2.5 concentrations were above some arbitrary threshold? It is unclear whether the specific days collected are representative of the Beijing region or were chosen specifically for other purposes. Why were not all days with available PM2.5 measurements analyzed? The authors must clarify in revision their methodology for selection of the days that were analyzed. The apparent arbitrary selection of studied days may affect interpretation of the results in Fig. S1 and S2, for example.

2. The authors state that the mean contribution of secondary particles to PM2.5 by the Si tracer method was 79.2% and refer to the reader to Fig. 4b (page 7, line 13). However, the bar graph in Fig. 4b shows a mean secondary concentration that appears to be closer to 70% rather than 80%. The authors should ensure the results they report in the text match those in the corresponding figures. Moreover, the clarity of Fig. 4a would be enhanced by labelling what boxes come from experimental measurements, what are estimations from inventories, etc.

3. The authors should enhance their discussion of uncertainties as well as clarify their

wording. For example, there are no lower error bounds on the values shown in Fig. 4. Error is discussed only for the annual average, but not for the seasonal or daily averages. The authors provide Table S1, which contains concentrations for chemical components indicative of secondary aerosols, but no similar table exists for the Si concentrations. Moreover, there are no uncertainties provided in Table S1. The manuscript would be substantially improved by a clear demonstration of a smaller uncertainty range for the Si approach. In addition, the authors suggest that the two methods being in close agreement is "proving the accuracy of the Si-dilution method" (page 8, lines 2-3). This is an odd statement as the authors spend most of the paper discussing the large uncertainties associated with the secondary tracer approach: could the agreement just be coincidental?

4. Lastly, this approach relies on Si not participating in secondary aerosol formation. The authors reference two previous studies on page 6, lines 7-9, to suggest that organosilicons do not contribute to aerosol formation. However, these studies are 15-25 years old and our understanding of Si chemistry has significantly advanced (as acknowledged in other portions of Section 3.3). In fact, volatile organic compounds like siloxanes are likely to become increasingly significant as aerosols of fossil fuel origin become less important over time (see: McDonald et al., "Volatile chemical products emerging as largest petrochemical source of urban organic emissions", Science, 2017, 359, 760-764, doi: 10.1126/science.aaq0524). In light of these trends, the authors should enhance discussion of the limitations of their approach. Is this approach only valid in heavily polluted urban environments with a large fossil fuel contribution (i.e. where Si contribution to secondary aerosol mass is minimal)?

---

## Author Comment (AC1) · 12 Feb 2019

Please see the attached ZIP file that includes the Response to Reviewer comments and the revised Manuscript and Supplement files.

Please also note the supplement to this comment:
https://www.atmos-chem-phys-discuss.net/acp-2018-914/acp-2018-914-AC1-supplement.zip

---

## Author Response (AR1)

**Response to Reviewers**

Dear Prof. Nizkorodov,

We really appreciate the comments that anonymous reviewers have made to improve the quality of our manuscript titled "Unraveling the role of silicon in atmospheric aerosol secondary formation: A new conservative tracer for aerosol chemistry" (acp-2018-914). We have now made substantial revisions to the manuscript according to the reviewers' comments. Each point raised by the reviewers has been carefully considered and replied beneath. The changes in the manuscript and Supplement are marked in red font. In this response letter, the comments from reviewers are in blue font and author's responses are in black font.

**Response to the Reviewer #1:**

*General comments: In the Earth's crust, silicon (Si) is ubiquitous and relatively inert element. The Si also existed in atmospheric aerosols. In this article, the author reports the role of silicon (Si) in the secondary aerosol formation. They found uncorrelated of the total mass of Si in PM$_{2.5}$ with the SA formation. Therefore, they suggested Si as a new conservative tracer for estimating the SA contribution to PM$_{2.5}$. Furthermore, the author also correlated the Si isotopic signatures with SO$_4^{2-}$, NO$_3^-$, NH$_4^+$, and SOC, then deduced the sources of these secondary components. This is a new method in estimating the secondary aerosol contribution to PM$_{2.5}$ by using the Si as a tracer. It's meaningful for the atmospheric chemistry. However, there still some comments need to be solved before the manuscript can be published. The author should consider that chemical species of secondary aerosols (e.g., NH$_4^+$, NO$_3^-$, SO$_4^{2-}$, SOC) as tracers are able to get the detail information of SA compare to use Si as a single tracer, what is the superiority of this method?*

**Response:** Thank you very much for your positive comments. The main objective of this work is: 1) to unravel the role of Si—a ubiquitous element in the environment—during the secondary aerosol formation process, which can enrich our knowledge on the Si in aerosol chemistry; 2) to suggest a new method for estimation of secondary aerosol contribution to PM$_{2.5}$ based on Si. We show some advantages of the Si-based method such as accuracy and simplicity (uses only a single tracer); however, we must stress that it does not mean to replace the traditional multi-tracer method, but adds a new alternative tool into the toolbox of aerosol chemistry research. The Si as a new conservative tracer can provide more information about the secondary aerosol formation process (e.g., SA contribution and potential sources of SA precursors). To properly describe the significance of this work and avoid misunderstanding, we have revised some related statements in the manuscript. Please see page 2 line 19; page 9 line 16-17; etc.

*Detail comments:*
*2.2 Sampling of PM$_{2.5}$ samples*
*(1) "the PM$_{2.5}$ samples were collected around Beijing on random haze days (n = 100) in 2013".*

*The author should give more detail information about the samples, such as the concentrations of PM$_{2.5}$, meteorological parameters, etc. These environmental conditions can help the reader to understand the latter results and discussion more clearly. These are also very important for the conclusion especially the contribution of SA in PM$_{2.5}$ during this article.*

**Response:** Thanks for your suggestion. We have added more detailed information about the samples including the concentration of PM$_{2.5}$ and all meteorological parameters (daily average air temperature, relative humidity, vapor pressure, wind speed, wind direction, atmospheric pressure, ground temperature, radiation, and photosynthetically active radiation) in the revised manuscript. Please see Table S1 in the Supplement.

*(2) What is the meaning of "random haze days (n=100)"? Can these samples represent the secondary formation of particles?*

**Response:** Thanks for your question. In this revision, we have corrected the confusing statement "random haze days" and added more detailed information about the sampling days. The sampling dates were selected based on the following considerations: 1) they covered every week of the year as soon as possible; 2) in each week, the day with relatively heavy haze weather was selected; 3) several haze episodes over consecutive days were also selected for investigating the secondary aerosol formation process. In this way, we thought that the sampling days should be able to reflect the haze pollution condition of the whole year. The detailed sampling dates are listed in Table S1 in the Supplement. From Fig. S1a, the yearly PM$_{2.5}$ profile based on the selected sampling days in 2013 was consistent with that reported previously (Lu et al., 2018), confirming the representative of the data. By the way, the "$n = 100$" has been corrected to "$n = 63$". Please see page 2 line 31-page 3 line 1; page 5 line 1-3.

*(3) The sampling sites should be given a brief introduction. For example, what the feature around the sampling sites? What the location of the sites?*

**Response:** We have added a detailed description about the sampling sites in the revised manuscript. The sampling site was located in an urban district of Beijing (116.4254 °E, 40.0481 °N) at ca. 20 m height above the ground and surrounded by office and residential buildings. Please see page 2 line 31-35.

**2.6 Uncertainty analysis in the estimation of secondary aerosols**

*(1) Emission Inventory may introduce significant uncertainties because of uncertainties in activity-related data and emission factors. At least, the authors should state clearly the effect of emission inventory so that the readers can judge by themselves. Furthermore, "assuming that the relative emission ratios of primary sources are steady" what's the meaning? The high temporally resolved emission inventory could be used in Table 1, other than the total emissions mass. For example, the monthly emission mass from emission inventory could be used other than yearly emission mass.*

**Response:** Thanks for your kind suggestion. First, we have enhanced the discussion on the uncertainties of emission inventory. According to your suggestion, we have clearly stated in the revised manuscript that emission inventory could indeed affect the uncertainty of the estimate result. Since the emission inventory used in this study (i.e., the MEIC database) does not include any error information of the data, to help the readers with understanding the effect of emission inventory, we show an example for the effect of emission inventory: if assuming the uncertainty of the emission inventory to be 5%, the uncertainty of the annual mean $f_{sec}$ in 2013 would be increased from 26.1% to 29.3%. Please see page 4 line 24-28. Furthermore, we have also alerted the readers the effect of the uncertainty of emission inventory in the Sect. 3.5. Please see page 7 line 21-24.

Second, we have removed the ambiguous statement "assuming that the relative emission ratios of primary sources are steady" from the manuscript. Please see page 4 line 25.

Third, we agree that a higher temporally resolved emission inventory would be better to obtain more accurate result. Unfortunately, the MEIC database can only provide yearly emission mass data. So, this possibility has been discussed in the manuscript but can only be achieved in future studies. Please see page 4 line 28-30 and page 7 line 3-4. Despite that, this should not hinder the readers from understanding the main concept of this work.

*From 3.1 to 3.2, the author correlated the Si with the secondary species in PM$_{2.5}$, the secondary precursors, and relative humidity. There are some comments about this part:*

*(1) In the part of Methodology, the author stated that random haze days n=100, in this part, n=63. What the meaning of "random haze days"? why these days were used to do the analysis? The author found the PM$_{2.5}$ concentration showed a clear seasonal trend. How was the representative of the data in the article? Have the author compared their results with references?*

**Response:** Thanks for your questions. As responded above, first, we have corrected the wrong sample number "100" to "63". Second, we have given the detailed principles for selecting sampling days in the revised manuscript: 1) they covered every week of the year as soon as possible; 2) in each week, the day with relatively heavy haze weather was selected; 3) several haze episodes over consecutive days were also selected for investigating the secondary aerosol formation process. The yearly PM$_{2.5}$ profile (Fig. S1a) was consistent with that reported previously (Lu et al., 2018). Thus, we thought that the sampling days should be able to reflect the haze pollution condition of the year. Please see page 2 line 31-page 3 line 1; page 5 line 1-3; and Table S1 in the Supplement.

*(2) All the samples were collected during haze days, why they don't collect some samples during clean days before these haze days? The comparison from clean days to haze days can help to give more information during the formation of SA.*

**Response:** Thanks for your suggestion. The detailed sampling dates are listed in Table S1 in the Supplement. They did include several clean days (e.g., 18/8/2013, 7/11/2013). In fact, we have collected samples from several haze episodes over consecutive days for investigating the secondary aerosol formation process (as shown in Fig. 4d-e and Fig. S6 in the Supplement). In this revision,

we have also added more discussion on specific haze episodes. Please see page 8 line 2-6 and line 24-34.

**3.4 Secondary aerosol estimate using the Si-dilution method:**

*(1) The uncertain caused by atmospheric transport should be considered in this study.*

**Response:** Thanks for your suggestion. In this revision, we have added a discussion on the potential effect of atmospheric transport on the estimate result. Here the PM$_{2.5}$ was assumed to mainly derive from local emission of primary particles and SA precursors. In fact, a portion of it might also come from adjacent cities/regions via atmospheric transport, which could cause uncertainties to the result. Therefore, using primary source information and emission inventory covering adjacent cities/regions in the calculation may also improve the accuracy of the result. Please see page 7 line 4-9.

*(2) Some special episodes should be analysis to support the views and better understand for the readers in this work.*

**Response:** Thanks for your suggestion. We have added the results and discussion for a special haze episode (Oct 16 to Oct 17 in 2013) and compared its results with the traditional method. Please see Fig. 4d-e and page 8 line 2-6.

**3.5 Comparison with the traditional method**

*The author stated that the "traditional method bring in huge uncertainties to the result". Then the author compared the Si-dilution method with the traditional method, "the results between the two methods were actually very close". Why?*

**Response:** Thanks for your comment. The "huge uncertainties" refers to the potential uncertainties of the traditional method that only considered the number of variables used in the calculation, as more variables may bring in more uncertainties in the calculation. While, "the results between the two methods were close" refers to the comparison of experimental estimate results in this specific case. Therefore, there should be no direct dependence between these statements.

Furthermore, to make the statements more accurate and avoid misunderstanding, we have substantially improved the discussion on the uncertainties. We do not claim that the present method gives less uncertainties than the traditional method, for the absence of the uncertainties of emission inventory makes it difficult to directly compare the uncertainties between the two methods. Please see page 7 line 30-33.

**3.6 Sources of secondary particle precursors revealed by Si isotopic signatures**

*The discussion in section 3.6 is too simple and less persuasive. The author should give more detail analysis of even one haze case process. Furthermore, there are many works have been done about the pollution case in the year 2013 in Beijing. The author should add more comparison between their results with the references.*

**Response:** Thanks for your suggestion. In this revision, we have added the analysis of two special

haze episodes in the Sect. 3.6 (i.e., Nov 30 to Dec 1, 2013 and Dec 6 to Dec 7 in 2013). We have analyzed all secondary species and precursors during these two haze episodes and discussed their correlation with the Si isotopic signatures of PM$_{2.5}$. The results were also compared with those in the literature. Please see page 8 line 285-295 and Fig. S6 in the Supplement.

**3.7 Implications for air pollution control policies**

*Huang's results were based on the observation during the high pollution events of 5–25 January 2013 at the urban sites of Beijing. The SA contribution obtained in this article was 88.7% for Jan 2013. This result deduced from just about five days daily data (6/1/2013, 11/1/2013, 18/1/2013, 25/1/2013, 29/1/2013), can these data have the representative for 20 days from 5-25 Jan 2013? The SA contribution obtained was 79.2%, what's the uncertainty of these results?*

**Response:** Thanks for your comment. First, we noted that the SA contribution in Jan 2013 obtained here showed some difference from that in Huang's paper. Such a difference might result from the different dates: as stated in the Sect. 2.2, the present work focused more on haze days, and Huang's paper selected 20 consecutive days. Therefore, the results obtained in this work should not be directly comparable with that in Huang's paper. This point has been specified in the revised manuscript. Second, we have added uncertainty for the data 79.2%. Please see page 9 line 2-4.

*Besides, there are some grammar and spelling mistakes. The format of some reference should be checked, such as line10 in section 3.3, Line12-13 in section 3.3.*

**Response:** We have carefully gone through the manuscript to avoid any grammar or spelling mistakes. The format of references have also been carefully checked and corrected. Please see page 6 line 1.

**Response to the Reviewer #2:**

*This manuscript (acp-2018-914) describes the use of Si to estimate the contribution of secondary aerosol to PM$_{2.5}$ in the urban environment in Beijing, China. The approach relies on Si content from PM$_{2.5}$ arising from primary sources only. Secondary aerosol concentrations are estimated by the inferred dilution of expected Si mass. This dilution is calculated by comparing measured Si concentrations in PM$_{2.5}$ to estimated values based on sources of primary emission. This approach is compared to another approach where concentrations of several secondary species (NH$_4^+$, NO$_3^-$, SO$_4^{2-}$, secondary organic carbon) are used as tracers for secondary aerosol production. The Si dilution approach gives a value in line but slightly higher than those given by the approach using chemical components associated with secondary aerosols. The authors argue the Si approach is more direct and has a lower uncertainty associated with it compared to the other approach. Moreover, the authors correlate secondary aerosol components in PM$_{2.5}$ (SO$_4^{2-}$, NO$_3^-$, NH$_4^+$, and secondary organic carbon) with Si isotopic composition and observe some correlations that suggest some sources for secondary aerosol.*

*This manuscript represents a new approach to estimate secondary aerosol formation and is within*

*the scope of Atmospheric Chemistry and Physics. However, several important issues, detailed below, must be carefully addressed by the authors before this manuscript is suitable for publication.*

***Comments:***

*1. The authors examined PM$_{2.5}$ collected on "random haze days" (see page 3, line 3; page 5, line 4). It is unclear what "random haze day" means. Were the analysis days randomly selected? Were the selected days the ones with the highest PM$_{2.5}$ concentrations, or days where PM$_{2.5}$ concentrations were above some arbitrary threshold? It is unclear whether the specific days collected are representative of the Beijing region or were chosen specifically for other purposes. Why were not all days with available PM$_{2.5}$ measurements analyzed? The authors must clarify in revision their methodology for selection of the days that were analyzed. The apparent arbitrary selection of studied days may affect interpretation of the results in Fig. S1 and S2, for example.*

**Response:** Thanks for your questions. In this revision, we have corrected the confusing statement "random haze days" and clarified the methodology for selection of the sampling days. The sampling dates were selected based on the following considerations: 1) they covered every week of the year as soon as possible; 2) in each week, the day with relatively heavy haze weather was selected; 3) several haze episodes over consecutive days were also selected for investigating the secondary aerosol formation process. The detailed sampling dates are listed in Table S1 in the Supplement. From Fig. S1a, the yearly PM$_{2.5}$ profile based on the selected sampling days in 2013 was consistent with that reported previously (Lu et al., 2018). Therefore, we thought that the selected days should be representative of the haze pollution condition of the whole year 2013 of the Beijing region. Please see page 2 line 31-page 3 line 1 and page 5 line 1-3.

*2. The authors state that the mean contribution of secondary particles to PM$_{2.5}$ by the Si tracer method was 79.2% and refer to the reader to Fig. 4b (page 7, line 13). However, the bar graph in Fig. 4b shows a mean secondary concentration that appears to be closer to 70% rather than 80%. The authors should ensure the results they report in the text match those in the corresponding figures. Moreover, the clarity of Fig. 4a would be enhanced by labelling what boxes come from experimental measurements, what are estimations from inventories, etc.*

**Response:** Thanks for your suggestion. First, we have corrected the mistakes in Fig. 4b to keep consistent with the text. Second, we have indicated the boxes in Fig. 4a that come from estimations from emission inventories with asterisks. This indeed makes the figure more understandable. Please see Fig. 4.

*3. The authors should enhance their discussion of uncertainties as well as clarify their wording. For example, there are no lower error bounds on the values shown in Fig. 4. Error is discussed only for the annual average, but not for the seasonal or daily averages. The authors provide Table S1, which contains concentrations for chemical components indicative of secondary aerosols, but no similar table exists for the Si concentrations. Moreover, there are no uncertainties provided in Table S1. The manuscript would be substantially improved by a clear demonstration of a smaller*

*uncertainty range for the Si approach. In addition, the authors suggest that the two methods being in close agreement is "proving the accuracy of the Si-dilution method" (page 8, lines 2-3). This is an odd statement as the authors spend most of the paper discussing the large uncertainties associated with the secondary tracer approach: could the agreement just be coincidental?*

**Response:** Thanks for your suggestion. In this revision, we have substantially improved the presentation and discussion on the uncertainties. First, we have added lower error bounds on the values in Fig. 4 as well as in Fig. S6 in the Supplement. The error information has also been given on seasonal or daily basis with some specific haze episodes. Please see Fig. 4c and 4e.

Second, we have added Si concentration on each sampling day in Table S2 in the Supplement (the old Table S1). The uncertainties have also been provided in Table S2 for all parameters including Si concentration and secondary species. How their uncertainties were obtained is given in the Supplementary experimental details. Please see Sect 1.3 and 1.5 in the Supplement.

Third, to make the statements more prudent and accurate, we have made substantial revisions to the discussion on the uncertainties. We do not claim that the present method gives less uncertainties than the traditional method, for the absence of the uncertainties of emission inventory makes it difficult to directly compare the uncertainties between the two methods. Please see page 7 line 25-34. Furthermore, in addition to the yearly and seasonal results (Fig. 4b-c), we have also added results for a special haze episode in the revised manuscript (Fig. 4e), which also indicated the consistency between the two methods. Thus, all results mentioned above suggested the accuracy of the present method. Please see Fig. 4 and page 8 line 2-6.

*4. Lastly, this approach relies on Si not participating in secondary aerosol formation. The authors reference two previous studies on page 6, lines 7-9, to suggest that organosilicons do not contribute to aerosol formation. However, these studies are 15-25 years old and our understanding of Si chemistry has significantly advanced (as acknowledged in other portions of Section 3.3). In fact, volatile organic compounds like siloxanes are likely to become increasingly significant as aerosols of fossil fuel origin become less important over time (see: McDonald et al., "Volatile chemical products emerging as largest petrochemical source of urban organic emissions", Science, 2017, 359, 760-764, doi: 10.1126/science.aaq0524). In light of these trends, the authors should enhance discussion of the limitations of their approach. Is this approach only valid in heavily polluted urban environments with a large fossil fuel contribution (i.e. where Si contribution to secondary aerosol mass is minimal)?*

**Response:** Thanks for your kind suggestion. We have enhanced the discussion on organosilicons in the revised manuscript. This part of discussion not only includes some early references but also cites some recent references on the relevant topic (e.g., McDonald et al., 2018; Wu and Johnston, 2017; Janechek et al., 2017). We quite agree that VOCs like siloxanes are likely to become increasingly significant in the SA formation as aerosols originating from fossil fuels may become less important over time. This point has been clearly indicated in the revised manuscript.

Furthermore, we have also clearly indicated the limitation of the method in the manuscript. Since the method is based on the dilution effect of Si in primary particles during the SA formation, it should be suitable for the environment where secondary Si production is minimal. Nevertheless, in our opinion, the method also keeps the flexibility of being modified to be applicable for the cases when the secondary Si production is not negligible. Since the estimation is based on the mass balance calculations (Sect. 2.5), it is possible to estimate the secondary Si production mass based on the airborne concentrations of organosilicons and hydroxyl radicals and include it in the mass balance calculation. Please see page 7 line 10-17.

Finally, we thank again the editor and the reviewers for your great efforts on improving the quality of this manuscript. We are looking forward to hearing your decision soon.

Thank you very much!

Best wishes,

Yours sincerely,

Qian Liu